# Assessing protected areas as climate refugia for threatened plant species in Britain

Freya R. Read[1]*, Rachel J. Warmington[2], Colin M. Beale[1,3]

1 Leverhulme Centre for Anthropocene Biodiversity, Department of Biology, University of York, York, United Kingdom, 2 Plantlife, Salisbury, Wiltshire, United Kingdom, 3 York Environmental Sustainability Institute, University of York, York, United Kingdom

* freyaread@hotmail.com

## Abstract

Climate change is causing the loss or movement of suitable habitats, forcing species to undergo range shifts. However, many may be unable to move to suitable locations, resulting in increased extinctions. Climate refugia, areas maintaining suitable conditions for species now and in the future, offer protection from climate change. These refugia are categorised as in situ, where species are currently present, and ex situ, where they are not. We aimed to identify climate refugia for rare vascular plant species in Britain and examined whether refugia with the highest number of suitable species share any common features. Using predicted 2080 distributions, we identified protected areas in Britain that act as in situ or ex situ refugia for 12 of Plantlife's focal plant species. This study revealed that many current habitats will become unsuitable by 2080, highlighting the urgent need to protect refugia. The limited availability of in situ refugia underscores the importance of ex situ refugia as potential habitats. Our findings indicated that protected areas with higher elevations, larger elevational ranges, higher latitude and longitude and larger area will provide climate refugia for a greater number of species. This information can be used to guide the selection and management of protected areas and identify receptor sites for species introductions.

## Introduction

Rising global temperatures are a major threat to biodiversity [1]. Climate change has put many species at an increased risk of extinction as it leads to the loss or movement of suitable habitats [2–4]. In order to survive, many species are required to shift their ranges [5,6], often moving to higher elevations and latitudes [7–9]. Projected rates of climate change are likely to exceed the ability of many species to move or adapt in time [3,10], leading to increased rates of extinction [11]. Landscape fragmentation by humans further limits species' ability to disperse and move to favourable environments [8,12,13]. While protected areas (PAs) around the world offer species refuge from human activities, there are concerns about whether their limited connectivity will prevent species from undergoing climate induced movement [13,14].

**Data availability statement:** The data and code used for this paper are accessible at: https://github.com/freya2121/climate-refugia.

**Funding:** The author(s) received no specific funding for this work.

**Competing interests:** The authors have declared that no competing interests exist.

Climatic refugia describe locations that maintain suitable conditions for species both now and in the future [15,16]. Such locations have high climate stability, enhancing the likelihood of population survival [15]. In situ refugia, which are refugia that are currently inhabited, have been shown to be limited in number, revealing that some species may lack any habitats that are likely to maintain climate suitability by 2070 [17]. It is therefore important to identify ex situ refugia: areas that have current and future suitability but are currently uninhabited by the species [16,17]. These areas are critical as they are potential relocation sites for species under threat. However, they are often inaccessible to species via natural dispersal, due to habitat fragmentation and species' limited dispersal abilities [8,17]. In cases where natural range shift is not possible, assisted colonisations may be the only option [3,18,19]. This is a type of conservation translocation that involves the intentional movement and release of a species outside of its current occupied range [2].

Results of the Botanical Society of Britain and Ireland's Plant Atlas (2020) reveal that 53% of native plants in the UK have declined since the 1950s [20]. This is partly due to the effects of climate change [20]. Pearce-Higgins et al [21] identified vascular plants and bryophytes as the groups with the highest proportion of species at risk from climate change in the UK, highlighting the urgent need for immediate action to protect our native plant species. Identifying in situ refugia for endangered plants pinpoints locations that require protection and conservation [17,22]. Previous work has assessed the availability of in situ and ex situ refugia for native plant species in Australia [16,17]. It is important to carry out similar research elsewhere to determine if protection of in situ refugia is sufficient or whether relocation to ex situ refugia is necessary for plant survival. Although Araújo et al. [23] identified in situ refugia for plants in Europe, they did not consider ex situ refugia. This emphasises the need for research in Britain that identifies both in situ and ex situ refugia in order to advise future conservation strategies.

This study focuses exclusively on PAs, in order to ensure that identified refugia are protected from both anthropogenic and climate-related threats [16]. Given the UK government's commitment to effectively protect and manage 30% of the land by 2030 [24], research guiding the selection of these PAs is crucial. While Baumgartner et al. [16] effectively identified in situ and ex situ refugia in Australia, the study did not explore common characteristics of these sites. Understanding the characteristics of climate refugia can help future conservation decision-making, giving us a better understanding of how we can protect species that are threatened by climate change [10,25,26]. Such results could help inform decisions on PAs, for example by identifying essential features for new PAs that maximise their effectiveness in a changing climate [27,28]. Climate change factors are currently overlooked in PA selection protocols, indicating a significant gap in conservation planning [23,26,28].

A combination of factors is likely to influence the suitability of PAs for species that are affected by climate change [29–31]. As global temperatures increase, PAs at higher latitudes and elevations in Britain are expected to become crucial refugia, offering cooler habitats that can support a large number of species [22,32–35]. Previous research has shown the importance of habitat size and diversity in PAs,

demonstrating that larger, more varied environments enhance species richness by providing diverse niches, resources, and microclimates [31,36,37]. Therefore, these factors are expected to increase the number of species that find climate refugia in such locations. If these variables are shown to be reliable indicators of climate refugia, they could serve as straightforward, assessable criteria to guide the management and establishment of new PAs.

We used species distribution models (SDMs) to identify in situ and ex situ climate refugia for 12 vascular plant species that Plantlife has identified as focal species for conservation action in Britain. We anticipate that these identified locations will be suitable habitats now and in the future (up to 2080). We then used this information to identify PAs that provide refugia for multiple species, in order to identify areas that are of high conservation importance. Next, we assessed whether the refugia with the highest number of suitable species share any common trends, testing the hypothesis that PAs with higher elevation, larger elevational ranges, higher latitude, lower longitude, larger area and higher habitat variation will provide climate refugia, both in situ and ex situ, for a greater number of species.

## Methods

### Selection of target species and protected areas

We examined the distributions of 12 native vascular plant species selected from Plantlife's focal species list (S1 Table). Plantlife is a UK-based conservation charity working to protect and restore wild plants and fungi, and currently manages 24 nature reserves across the UK [38]. Their focal species list is comprised of 30 threatened species: those located within their nature reserves; groups of species identified as important and/or under regarded; species which are the focus of multi partner projects; and those where recovery methods have been trialled and are now being implemented at scale. We excluded eighteen species that were recorded in fewer than five hectads to ensure sufficient data for modelling [21,27]. While this reduces the number of species analysed, the remaining 12 span a wide range of habitat types, levels of rarity and geographic distributions across the UK.

Species included:

- *Adonis annua* L. - Pheasant's-eye

- *Turritis glabra* L. - Tower Mustard

- *Carex ericetorum* Pollich- Rare Spring-sedge

- *Cerastium alpinum* L. - Alpine Mouse-ear

- *Dryas octopetala* L.- Mountain Avens

- *Galeopsis angustifolia* Ehrh. ex Hoffm. - Red Hemp-nettle

- *Juniperus communis* L. - Common Juniper

- *Mertensia maritima* (L.) Gray – Oysterplant

- *Bistorta vivipara* (L.) Delarbre – Alpine Bistort

- *Ranunculus tripartitus* DC. - Three-lobed Crowfoot

- *Silene conica* L. - Sand Catchfly

- *Spiranthes romanzoffiana* Cham. - Irish Lady's-tresses

We included all National Nature Reserves (NNR) and Sites of Special Scientific Interest (SSSI) in Scotland, England and Wales as potential refugia. These sites comprise the main PA estate in Britain [27]. Although the Natura 2000 network, including Special Areas of Conservation (SACs) and Special Protection Areas (SPAs), is significant, we did not include it

due to its frequent overlap with SSSIs [27]. The Natura 2000 sites that are not included in SSSI are predominantly marine areas, which are not relevant to this study [39,40]. Data on geographical boundaries, longitude, latitude and the area of the included PAs were provided by Natural England, DataMapWales and NatureScot.

## Current and future species distributions

Data on the current plant distributions were from the period 1970–86 from the Botanical Society of the British Isles Plant Atlas [39]. This data was recorded in 10 km² grid distributions [39]. The decision to use data from this period of the plant atlas, despite newer records existing, aligns with Critchlow et al.'s [27] choice for their distribution models. Whilst climate has changed dramatically over the past 40 years, this data was collected during a period of high effort to record distributions across the UK and precedes most of the modern climate change. This combination of quality data from a period when species were more likely to be in equilibrium with climate, makes our choice of dates appropriate.

Information on the predicted future distributions for the plant species in 2080 was from species distribution models (SDMs) carried out by Critchlow et al. [27] and full details can be found there. Briefly, these models are spatially explicit Bayesian hierarchical models [40] that integrate a spatially explicit generalised additive model including four bioclimatic variables and a model that accounts for variation in observer effort. Future projections modelled climate suitability using 11 regional climate model ensembles under the IPCC's A1B medium emission scenario [41].

We refined the modelled distribution to ensure that suitable areas for the plants were located only in their preferred habitat types (S1 Text). Overlapping these suitability maps with PA outlines (NNR and SSSIs), we computed the maximum suitability value for each PA.

We assigned a site suitability score to each location, ranging from 0 (unsuitable) to 1 (highly suitable), based on SDM outputs. To establish a baseline of suitability, we defined a suitability threshold. For each species, we calculated the 10th percentile of suitability scores of locations where the species was actually recorded during the period 1970–86, assuming that any suitable habitat with similar climate could also be managed to sustain the target species. We computed maximum suitability in both current and future datasets within each PA and applied the suitability threshold to identify PAs with current and future suitability. Finally, we overlaid current occupancy to these PAs in order to distinguish between in situ and ex situ refugia.

## Analysis of climate refugia variables

In order to identify variables that affected the number of species for which each PA provided refugia we collated data on latitude, longitude, mean elevation, elevational range, area and number of habitat types within the PA.

Information on latitude, longitude and area was computed from the PA boundaries. We determined the elevation of PAs using the 90m resolution GR SRTM Digital Elevation Model [42], accessed from Edinburgh DataShare. We calculated both the mean elevation and the elevational range within each PA. To determine the number of unique habitat types in each nature reserve, we used the UKCEH Land Cover Map [43] and counted the number of distinct habitat types within each PA.

We log-transformed mean elevation, elevational range and area before analysis to mitigate significant skew. We standardised all variables to a mean of zero and standard deviation of one.

To examine the relationship between the number of target species for introduction and the spatial predictors, we implemented a spatially-explicit Bayesian generalised linear mixed model with a Poisson distribution and log link using the glmmfields package [44]. All data analysis was conducted in R version 4.3.1 [45]. We used the packages tidyverse and tidyterra for data manipulation and visualisation [46,47]. For spatial analysis, we used terra and raster [48,49] Administrative boundaries used in figures were obtained from the GADM database [50] using the geodata package [51].

                                                                                  

## Results

For the 12 studied plant species we found that only 41.1% of their currently occupied habitats are projected to remain suitable in the future (Table 1). *S. romanzoffiana* is the only species predicted to have no overlap between its current and future distributions, although 27 potential introduction sites have been identified (Table 1). Ex situ refugia outnumber in situ refugia (Table 1), with 72% of identified refugia being currently unoccupied (Table 1). There is an average of 349.5 ex situ refugia for the 12 modelled species.

The spatial distribution of both in situ and ex situ refugia across Britain appears to have similarities, predominantly in upland and northern regions (Fig 1). Information on suitability for each species and each PA is provided in supplementary information (S1 Fig).

We found that the number of species sharing either in situ or ex situ refugia is positively associated with mean elevation (Figs 2A, 2B and 3), elevational range (Figs 2C, 2D and 3) and the area of PAs (Fig 2E, 2F and 3). We found evidence that more species found ex situ refugia in more northerly PAs (Figs 3B and S2B) but this was not significant for in situ refugia (Figs 3A and S2A). Additionally, the number of species with in situ and ex situ refugia in PAs were positively correlated with the longitude of the PAs (Figs 3 and S2C). There was no significant correlation between the number of habitat types within PAs and the number of species with either in or ex situ refugia in a PA (Figs 3 and S3). Parameter estimates and statistical support for these relationships in our spatially explicit models are provided in Tables 2 and 3.

## Discussion

### In situ and ex situ refugia

We found climate change threatens the future habitat suitability within Britain for the 12 species studied, predicting that on average, only 46% of the PAs they currently occupy will retain a suitable climate into 2080. Furthermore, all species are predicted to lose suitability in some of the PAs where they reside. Such general patterns are broadly supported by the wider literature. For example, a large study by Araújo et al. [23] examined the distributions of 1,200 plant species across Europe and found that 58% are expected to lose suitability in some of the PAs where they reside. The lower percentage

Table 1. The number of PAs classified as occupied, at risk, in situ refugia and ex situ refugia for each of the 12 focal species.

| | Occupied | At Risk | % At Risk | In Situ Refugia | Ex Situ Refugia |
|---|---|---|---|---|---|
| *A. annua* | 192 | 110 | 57.3 | 82 | 751 |
| *T. glabra* | 295 | 235 | 79.7 | 60 | 283 |
| *C. alpinum* | 63 | 9 | 14.3 | 54 | 109 |
| *C. ericetorum* | 200 | 190 | 95.0 | 10 | 202 |
| *D. octopetala* | 87 | 30 | 34.5 | 57 | 40 |
| *G. angustifolia* | 545 | 322 | 59.1 | 223 | 1,635 |
| *J. communis* | 1,482 | 711 | 48.0 | 771 | 654 |
| *M. maritima* | 45 | 21 | 46.7 | 24 | 45 |
| *B. vivipara* | 402 | 100 | 24.9 | 302 | 249 |
| *R. tripartitus* | 138 | 109 | 79.0 | 29 | 112 |
| *S. conica* | 148 | 102 | 68.9 | 46 | 87 |
| *S. romanzoffiana* | 13 | 13 | 100.0 | 0 | 27 |
| **Average** | 300.8 | 162.7 | 58.9 | 138.2 | 349.5 |

"Occupied" are PAs where the species is currently present, "at risk" are PAs where the species is currently found but climatic suitability is expected to be lost, "in situ refugia" are PAs where the species is currently found and that should remain suitable in the future, and "ex situ refugia" are PAs where climate and habitat are broadly suitable in both time periods but the species is absent. Additionally, the percentage of currently occupied PAs that are at risk is included.

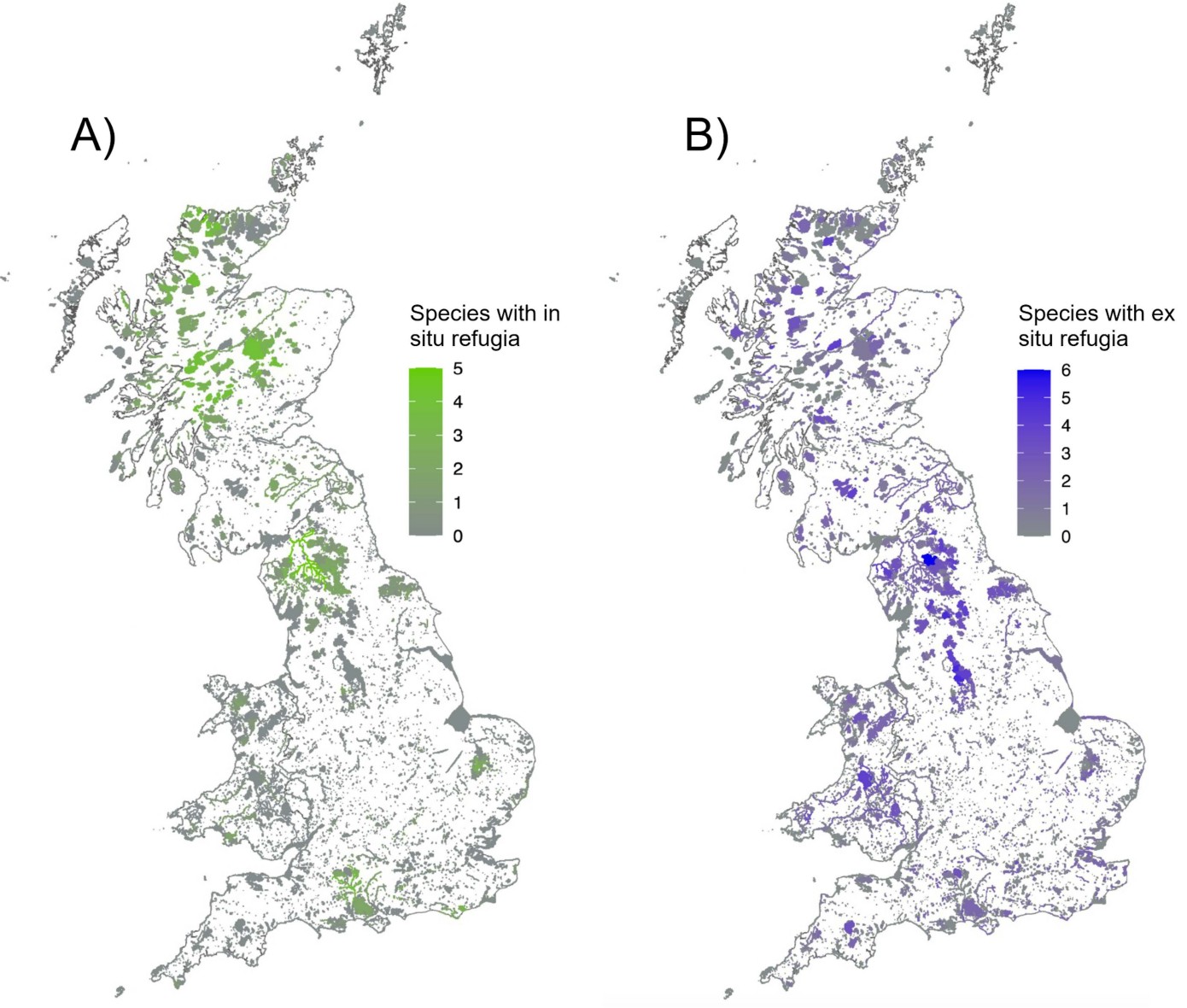

**Fig 1. Maps illustrating the number of species finding (A) in situ and (B) ex situ refugia in each NNR or SSSI across England, Scotland and Wales.** Shading intensity corresponds to the number of species within each site finding refugia within the PA.

they reported likely reflects the wider range of species examined, in contrast to our study's focus on 12 species with limited distributions. Additionally, by using European data, Araújo et al.'s [23] study encompasses a more complete representation of the species' natural ranges. Expanding our SDMs to include broader distribution data could reveal additional suitable locations for the species [21]. Both these results show the crucial need to identify the scarce in situ refugia for the effective conservation and management of species at risk. However, they also show that protecting current in situ refugia will not be enough to ensure the survival of species.

The limited number of in situ refugia means it is also important to identify ex situ refugia for threatened species [16,17]. We found approximately 72% of the sites identified as suitable are currently unoccupied by the species, highlighting a

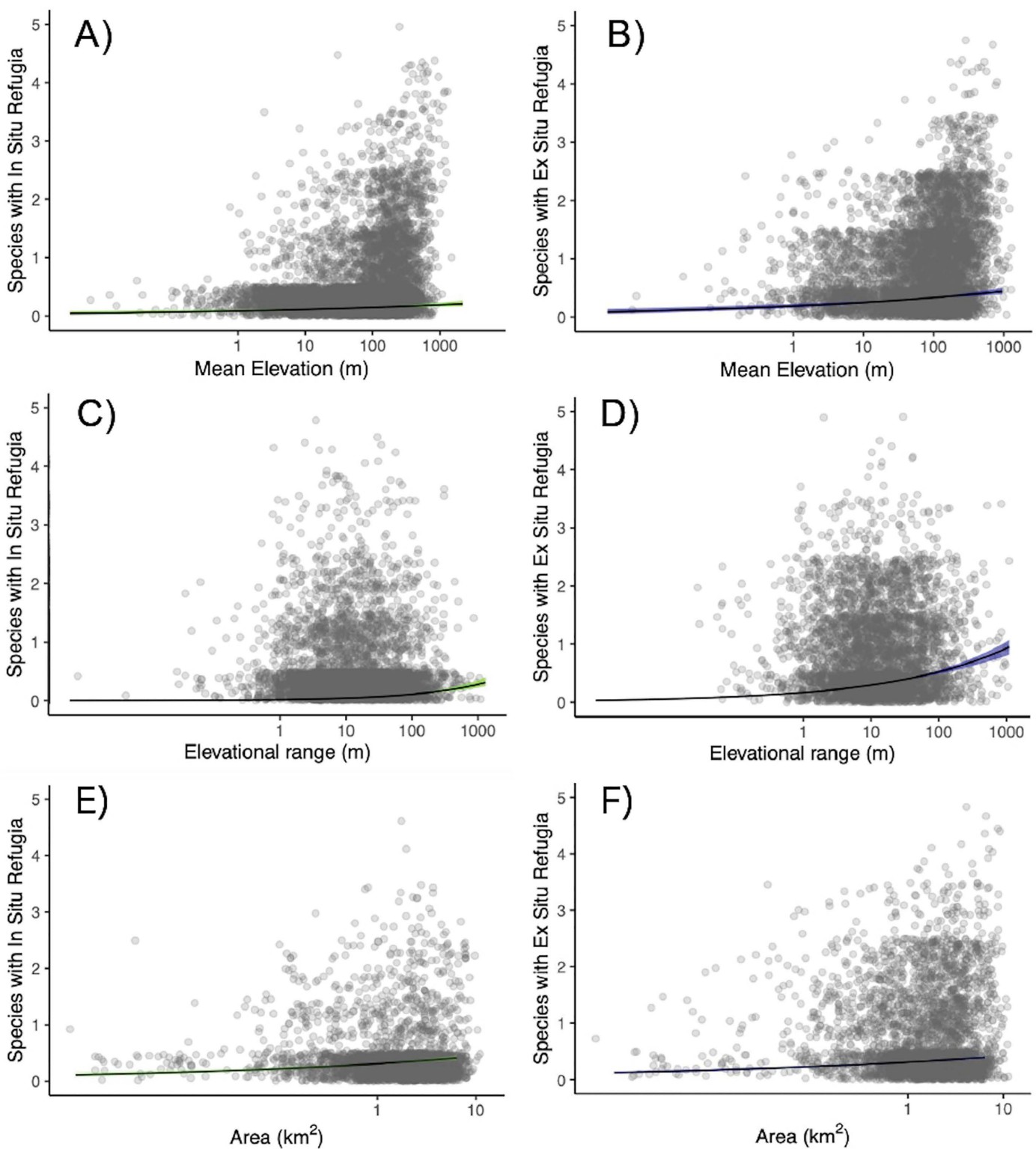

**Fig 2. Effect plots illustrating the relationship between PA covariates and the estimated number of species (sum of occupancy probabilities) finding refugia within the PA.** The covariates included are mean elevation **(A and B)**, elevational range (C and D) and area **(E and F)**. Figures on the left show the relationships with the number of species with in situ refugia, while those on the right depict species with ex situ refugia in each PA. Lines of

best fit derived from a Bayesian generalised linear mixed model are represented with 95% credible intervals (green for in situ refugia and blue for ex situ refugia). All graphs demonstrate a significant positive correlation.

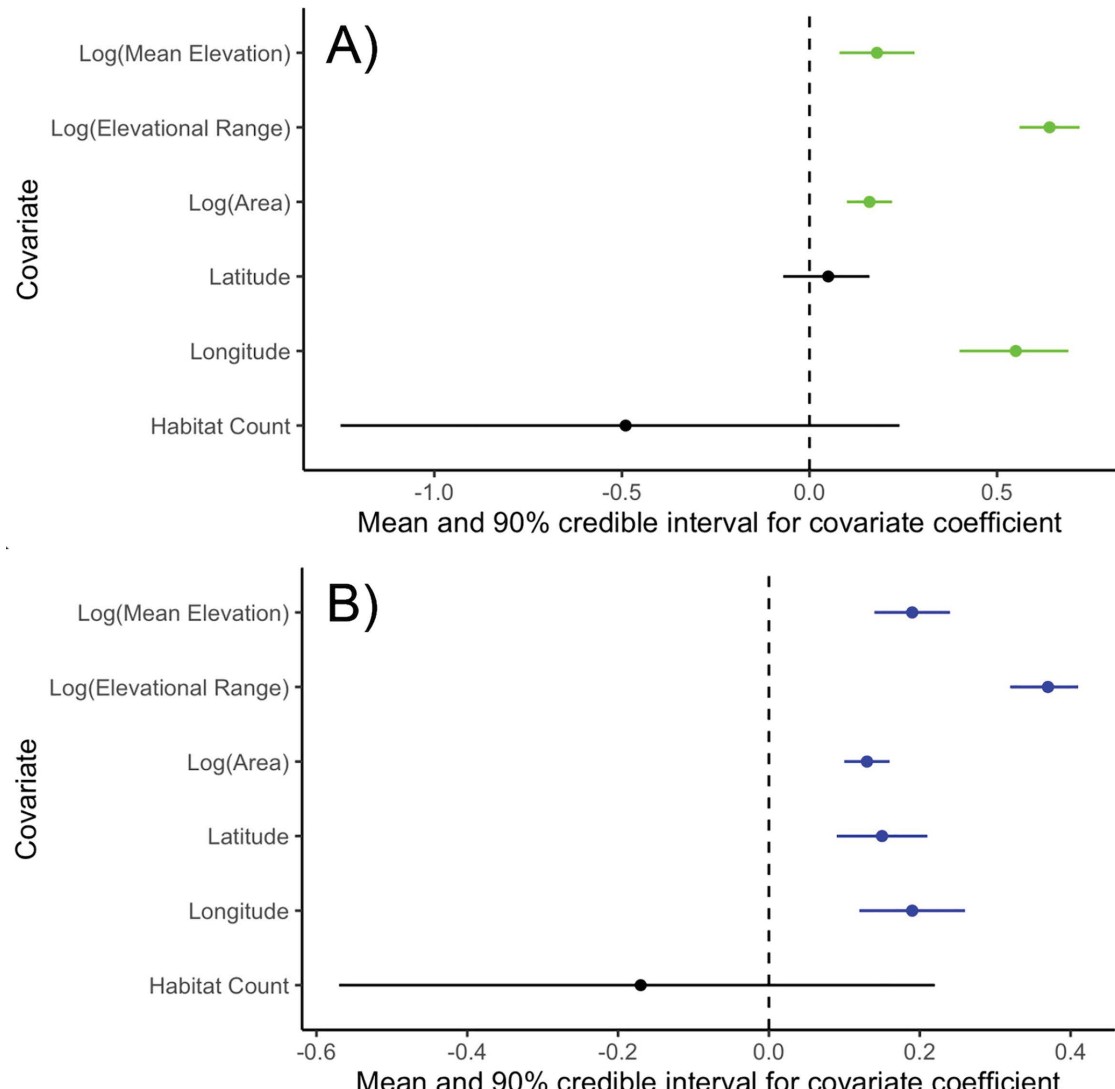

**Fig 3. Forest plot from the Bayesian generalised linear mixed model examining how variables affect the number of species with (A) in situ refugia and (B) ex situ refugia.** For each variable, the plot shows the mean effect, with error bars showing 95% credible intervals. Variables are indicated to have a significant positive effect if their credible intervals are above zero, a significant negative effect if their credible interval is below zero and no effect if the credible interval overlaps zero.

significant number of potential receptor sites for the assisted colonisation of threatened species. For the 12 modelled species, there is an average of 233 candidate sites for introductions. Some species may be able to shift and disperse to these habitats naturally, but given the fragmented nature of Britain's PAs and the already narrow distributions of the focal species, this is unlikely [17,52]. Climate and habitat suitability alone cannot fully describe the exact requirements of rare

**Table 2. The results of a Bayesian generalised linear mixed model to examine parameters associated with the number of species with ex situ refugia.**

| Parameter | Mean | SD | Lower | Upper |
|---|---|---|---|---|
| Intercept | −3.01 | 0.31 | −3.64 | −2.41 |
| Log(Mean Elevation) | 0.18 | 0.05 | 0.08 | 0.28 |
| Log(Elevational Range) | 0.64 | 0.04 | 0.56 | 0.72 |
| Log(Area) | 0.16 | 0.03 | 0.10 | 0.22 |
| Latitude | 0.05 | 0.06 | −0.07 | 0.16 |
| Longitude | 0.55 | 0.07 | 0.40 | 0.16 |
| Habitat Count | −0.49 | 0.38 | −1.25 | 0.24 |

The table includes the mean, standard deviation (SD) and 95% credible intervals (Lower and Upper bounds).

**Table 3. The results of a Bayesian generalised linear mixed model to examine parameters associated with the number of species with in situ refugia.**

| Parameter | Mean | SD | Lower | Upper |
|---|---|---|---|---|
| Intercept | −1.17 | 0.16 | −1.5 | −0.86 |
| Log(Mean Elevation) | 0.19 | 0.03 | 0.14 | 0.24 |
| Log(Elevational Range) | 0.37 | 0.02 | 0.32 | 0.41 |
| Log(Area) | 0.13 | 0.02 | 0.10 | 0.16 |
| Latitude | 0.15 | 0.03 | 0.09 | 0.21 |
| Longitude | 0.19 | 0.04 | 0.12 | 0.26 |
| Habitat Count | −0.17 | 0.20 | −0.57 | 0.22 |

The table includes the mean, standard deviation (SD) and 95% credible intervals (Lower and Upper bounds).

plants, which may require specific habitat management. Therefore, these analyses are only broad indicators of potential areas where application of appropriate management, combined with introductions might be successful.

For some species, assisted colonisations may be the only option to ensure their survival. Our results indicate that *S. romanzoffiana* has no in situ refugia. Previous research looking at this on a broader scale found that 7% of studied plants were projected to have no degree of overlap in their current and future distributions [23]. *S. romanzoffiana* has 27 PAs with possible introduction sites (ex situ refugia), offering opportunities for the survival of the species in Britain. These sites should be considered as candidate sites for translocation, requiring additional analysis of the sites and the species-specific conservation measures to ensure success.

## Features of climate refugia

Although we focused on 12 species, the results of our study highlight general trends in the location of suitable refugia for species at risk because of climate change. Higher elevation, larger elevational ranges, higher latitude and longitude, larger area and higher habitat variation are all important factors that increase the suitability of a site for reintroductions. These trends confirm existing guidance for adapting PA design to climate change [32] as well as wider studies of climate refugia [53] and can be used to advise the management of current PAs and inform decisions about the location of future PAs. They are also likely to remain significant factors beyond the modelled future climate for 2080 as global temperatures continue to rise.

Our results support the hypothesis that the greater the mean elevation of PAs, the more species there are that share refugia. This reflects the well-known upward shift in suitable climates for organisms, driven by rising global temperatures

[34,35]. There is substantial evidence that these elevational shifts are already occurring around the world, for example Lenoir et al. [35] observed a significant shift in elevation of forest plant species in Western Europe, averaging 29 metres per decade, and upward shifts in alpine plant distributions in the UK have been known since 2006 [54]. In the UK, PAs are currently disproportionately located in areas of high elevation [27,55]. Critchlow et al. [27] suggests that if the predicted shifts in species distribution due to climate change occur, the number of species represented in PAs in the UK is likely to increase. However, Alexander et al., [35] suggest that the altitudinal shifts of plants are lagging behind climate trends. This highlights the need for conservation strategies that not only recognize the importance of high-elevation PAs but also address the challenges species face in reaching these refugia.

We found a positive association between elevational range and the number of species sharing refugia within these areas. Similarly, Lawrence et al. [29] found a strong association between topographic diversity and increased climate change resilience of PAs worldwide [29]. This topographic variability results in high environmental diversity, with fine-scale temperature variations creating valuable micro-habitats [29,56]. This lowers the risks of population decline under climate change because habitats with varied climates are available for species within close proximity, and within the same PAs [57]. Bellis et al. [58] estimate a decrease of 1°C that occurs with a 167 m altitude increase is equivalent to a 145 km change in latitude [58]. A varied topography therefore allows species to meet their climatic needs with minimal migration, allowing them to undergo range shifts without leaving the confines of the PA [29,31,59]. Previous research has suggested that there is a higher level of threat associated with flat terrain under climate change as species have to move further to remain in suitable conditions [56,57,60]. Our research emphasises the importance of maintaining PAs that span elevational gradients in order to improve the effectiveness of the British PA network.

We generally found more species with ex situ refugia in more northern PAs. This fits with previous studies that have shown PAs in high latitude regions are likely to retain particularly high area proportions of climate conditions [61]. It further matches global patterns indicating that many species have already shifted their ranges poleward [34,62,63] a pattern well documented in the UK where 84% of examined species have migrated northwards [63]. Bellis et al. [58] and Temunović et al. [15] have found that globally, selecting translocation sites closer to the poles could reduce climate change impacts on the populations. More surprisingly, we did not find a correlation between the number of species with in situ refugia and the latitude of the site. While our findings do not support this correlation, they also do not contradict the existing research that indicates a positive relationship. The lack of significance in our data may reflect the preponderance of grassland and low-intensity cropland species in our list, which have a predominantly southern distribution in the UK. Additionally, the relatively low number of species included in our study may also contribute to this lack of significance. Overall, latitude is an important factor for identifying suitable plant conservation sites, and conservation efforts should support the species' northward migration in response to climate change.

We found a larger number of species finding refugia at higher longitudes. This differs from our hypothesis where we expected to find a larger number of species finding refugia in the West due to its cooler and wetter conditions [22,33]. However, the high number of species in the East suggests that other factors are important in determining where species find refugia. This eastern pattern may partly reflect the particular species that passed the inclusion threshold, which are those with sufficient occurrence records to be reliably modelled. Many of these species are associated with arable land, calcareous grasslands, or inland rock habitats, which are more prevalent in eastern and southeastern Britain, creating a natural eastward bias in suitable habitat availability. Species meeting this threshold may also be biased toward regions with high observer effort. The southeast has historically had better coverage, especially around major towns and cities [20,39]. Research in other countries has also shown a relationship between longitude and suitability [64], but this relationship is likely to vary between countries with differing longitudinal climatic and environmental gradients [65].

Our results support our hypothesis that the size of PAs is positively correlated with the number of species that share refugia within these locations. This is a widely accepted understanding that larger PAs are preferred due to greater habitat diversity, reduced edge effect, improved connectivity and greater capacity to support larger wildlife populations [31,36,66].

For example, Volenec and Dobson [66] found that reserve size is a significant predictor of plant species richness. Our study found a greater number of species with refugia in larger PAs, possibly because extensive and likely more diverse habitats can support a wider range of species. It is also probable that larger PAs are more likely to support species during climate change as they will allow range shifts within the PA, whereas small PAs are less likely to maintain areas with similar climatic conditions in the future [8]. This means that translocated species are anticipated to find suitable climates within the large PAs even beyond the modelled future climates in 2080.

Although many papers state the importance of larger PAs, Kendal et al. [67] argue that small PAs should not be ignored, as this can sometimes lead to the disregard of critically endangered species that are restricted to small PAs. However, for the purpose of species introductions or reintroductions, larger habitats are generally preferred. This is because they are able to support larger population sizes and facilitate necessary population expansions and range shifts [22]. The majority of Britain's PAs are currently very small, with most SSSIs being under 1 km$^2$ [27]. It is therefore important to increase the size of Britain's PAs to improve habitats for wild plants, and to ensure these habitats stay suitable in the future.

Our analysis found no correlation between the diversity of habitat types in PAs and the number of species sharing refugia within these locations. This is surprising, as more diverse land cover types within PAs would be expected to support a greater variety of plant species [37]. A possible reason is that the majority of our focal species are highly specialist, mainly adapted to grassland (notably calcareous and acidic) and inland rock habitats. Therefore, the number of our focal species that find refugia within a site relies more on the presence of these specific habitats, rather than the number of habitat types. Deák et al. [68] found that even specialised habitats show within-habitat variation caused by topographic heterogeneity and distinct abiotic conditions. Therefore, a more detailed fine-scale analysis of these habitats would likely reveal a correlation between microenvironmental variation and biodiversity [68,69]. This shows that even within specialised habitats, microenvironmental variation can enhance biodiversity, underscoring the importance of habitat variations and topographic diversity in promoting species diversity within PAs. The focal species of our study currently occupy a very small range within their broader land use category, so it may be this microenvironmental variation that is important for the presence of each species. Therefore, conservation strategies should prioritise the study and protection of these microhabitats to ensure the persistence of these rare species.

## Limitations

There is a possibility that our study overestimated the range sizes of these species due to the low spatial resolution of the SDMs used, at 10 km$^2$. Previous research has found that this is a common limitation of SDMs [28], especially as there is often a lack of high resolution climate variables [70]. To enhance SDMs, both Rowden et al. [71] and Lembrechts et al. [72] recommend employing high-resolution data. When this is not available, lower resolution SDMs remain valuable for conservation, but detailed assessments will be required to identify specific suitable locations for the translocation of species outside their original habitats, which would be necessary regardless [73].

Another limitation of the SDMs used is that they are based solely on the species distributions in Britain, rather than incorporating global distribution data. Therefore, they potentially underestimate the realized niche of the species. However, this means that any identified refugia are still very likely to be refugia. This doesn't prevent us from identifying suitable areas but means that the species may also survive in additional locations not captured by the model. The spatially explicit modelling approach used in this study explicitly seeks to minimise these effects, although it cannot eliminate them completely.

Since the SDMs we used did not account for microclimates and are based on synoptic temperature conditions that may not accurately reflect the temperature experienced by plants, they may miss small sites with suitable climate [25,72]. Incorporating higher-resolution environmental or topographic data in future analyses would strengthen these assessments. Tools such as climate data loggers, remote sensing and fine-scale digital elevation models can help identify microhabitats that contribute to local thermal buffering [25], improving our understanding of how protected areas function

as climate refugia. However, the lack of higher resolution data in this study is of limited concern as microrefugia are not sensible locations for species introductions, due to limited habitat size and limited possibility of expanding range [15]. It is also possible that species identified as not having suitable habitats in the future may still occur in small areas where microclimates remain suitable [17], so complete removal of at risk species is not advised. These locations are likely to be too small to guarantee species survival in the long term but could serve as a source for future translocations.

Our use of current habitat maps to identify suitable locations in 2080 does not account for potential habitat shifts due to climate change, as habitat types may change or move over time. However, combining current habitat maps with future climate projections helps to highlight areas that are most likely to retain both suitable habitats and climates. Additionally, in the UK large-scale transitions in land cover are more likely to be the result of a combination of human land use change and climate change. While integrated scenarios of climate and land use change are becoming available, they are not yet suitably detailed to enable modeling. Over the time periods we looked at, most models estimate more land covers will remain as they are rather than transition [74], making our results informative.

## Conclusion

Our examination of the current and future habitat suitability of 12 plant species within Britain highlights the significant threat posed by climate change. Our findings reveal that a high number of species' current habitats will become unsuitable by 2080. The limited number of in situ refugia means there is an urgent need to identify and protect ex situ refugia. Ex situ refugia provide suitable new locations for species that have limited in situ refugia, however the fragmented nature of Britain's PAs is a significant barrier to species' natural relocation [52,55]. Translocations can be used to relocate species to suitable habitats, a tool that will become increasingly important as global temperatures continue to rise [2,75].

Our study provides insights into the general characteristics of climate refugia. We found PAs with higher elevations, larger elevational ranges, higher latitude and longitude and larger area will provide climate refugia for a greater number of species. This information can be used to select new PAs, advise the management of current PAs and to identify receptor sites for the introduction of species. Selecting sites with these easily identifiable characteristics enhances their potential to remain viable habitats for species, even beyond the modelled future climates. Globally, countries are increasing efforts to expand PAs [76]. Considering climate change in PA selection protocols [23,28], is crucial for ensuring the resilience and effectiveness of PAs.

Our research has highlighted the significance of not just cooler temperatures but also the importance of a range of temperatures within a PA. Research using a high spatial resolution, using climate data loggers and remote sensing, is important to further refine translocation priorities [25]. Additionally, these techniques can be used to learn more about the specific habitat requirements of target species, therefore facilitating the development of species-specific conservation strategies.

Overall, our study demonstrates the existence of climate refugia within Britain, offering hope for the survival of species threatened by climate change. It underscores the critical need to protect these refugia and to develop conservation strategies for species that are unable to migrate independently to these suitable locations. These actions are crucial for preserving biodiversity not only within Britain but globally. By ensuring that threatened species can access suitable refugia, we will be able to significantly reduce biodiversity loss due to climate change.

## Supporting information

**S1 Table. Summary information of the species included in this study.** This includes their scientific and common names, species descriptions [1], broad habitat types [2], and corresponding UK CEH land cover classes and identifiers [3]. (PDF)

**S1 Text. Refinement of species distribution based on habitat suitability.**
(PDF)

**S1 Fig. Maps illustrating the distribution of in situ refugia, ex situ refugia and areas at risk in each NNR and SSSI across England, Scotland and Wales.** Green represents PAs where the species have in situ refugia, blue denotes ex situ refugia, and red indicates areas where the species is at risk. PAs that do not fall into these categories are not shown. (PDF)

**S2 Fig. Effect plots illustrating the relationship between PA covariates and the estimated number of species (sum of occupancy probabilities) finding refugia within the PA.** The covariates included are latitude (A and B) and longitude (C and D). Figures on the left show the relationships with the number of species with in situ refugia, while those on the right depict species with ex situ refugia in each PA. Lines of best fit derived from a Bayesian generalised linear mixed model are represented with 95% credible intervals (green for in situ refugia and blue for ex situ refugia). All graphs show a significant positive correlation, except for the one between latitude and the number of species with in situ refugia, which shows no correlation. (PDF)

**S3 Fig. Bar graphs illustrating the number of habitat types within each PA alongside the number of species with refugia.** Graph (A) illustrates species with in situ refugia, while graph (B) depicts those with ex situ refugia. Null data points were omitted. Neither graph identified a correlation between the number of species with refugia and the number of habitat types. (PDF)

**S2 Text. Plantlife nature reserves case study.** (PDF)

**S2 Table. Presence and suitability of focal species for Plantlife's reserves, arranged by reserve.** The "Suitable_ score" column is the probability of the reserve being suitable now and in the future. The "Suitable" column indicates if this value is above the suitability threshold. The "Presence" column indicates if this species is already there. "Introduction" indicates if the reserves are suitable for the species but the species is not currently present, indicating opportunities for species introduction. "At Risk" indicates species that are present but do not have both current and future suitability. (PDF)

**S3 Table. Presence and suitability of focal species for Plantlife's reserves, arranged by reserve.** The "Suitable_ score" column is the probability of the reserve being suitable now and in the future. The "Suitable" column indicates if this value is above the suitability threshold. The "Presence" column indicates if this species is already there. "Introduction" indicates if the reserves are suitable for the species but the species is not currently present, indicating opportunities for species introduction. "At Risk" indicates species that are present but do not have both current and future suitability. (PDF)

## Acknowledgments

We would like to thank Elva Robinson for their comments on the draft of this article, and the many recorders who have contributed data towards the Botanical Society of the British Isles Plant Atlas. We are also grateful for the information and advice provided by Dr Elizabeth Cooke and colleagues at Plantlife.

## Author contributions

**Conceptualization:** Freya R. Read, Rachel J. Warmington, Colin M. Beale.

**Data curation:** Freya R. Read.

**Formal analysis:** Freya R. Read, Colin M. Beale.

**Investigation:** Freya R. Read.

**Methodology:** Freya R. Read, Rachel J. Warmington, Colin M. Beale.

**Supervision:** Rachel J. Warmington, Colin M. Beale.

**Validation:** Colin M. Beale.

**Visualization:** Freya R. Read.

**Writing – original draft:** Freya R. Read.

**Writing – review & editing:** Freya R. Read, Rachel J. Warmington, Colin M. Beale.

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
