## [Decision Letter · Decision Letter 0]

21 Oct 2025

Dear Dr. Read,

plosone@plos.org. A rebuttal letter that responds to each point raised by the academic editor and reviewer(s). You should upload this letter as a separate file labeled 'Response to Reviewers'.A marked-up copy of your manuscript that highlights changes made to the original version. You should upload this as a separate file labeled 'Revised Manuscript with Track Changes'.An unmarked version of your revised paper without tracked changes. You should upload this as a separate file labeled 'Manuscript'.

We look forward to receiving your revised manuscript.

Kind regards,

Francesco Boscutti

Academic Editor

PLOS ONE

Journal Requirements:

2. Please amend your authorship list in your manuscript file to include author Freya Rebecca Read, Rachel Warmington, Colin Beale .

3. Please amend the manuscript submission data (via Edit Submission) to include author Freya R. Read, Rachel J. Warmington, Colin M. Beale.

5. We note that Figure(s) 1, S1 in your submission contain [map/satellite] images which may be copyrighted. All PLOS content is published under the Creative Commons Attribution License (CC BY 4.0), which means that the manuscript, images, and Supporting Information files will be freely available online, and any third party is permitted to access, download, copy, distribute, and use these materials in any way, even commercially, with proper attribution. For these reasons, we cannot publish previously copyrighted maps or satellite images created using proprietary data, such as Google software (Google Maps, Street View, and Earth). For more information, see our copyright guidelines: http://journals.plos.org/plosone/s/licenses-and-copyright.

a. You may seek permission from the original copyright holder of Figure(s) 1, S1 to publish the content specifically under the CC BY 4.0 license.

Additional Editor Comments :

In general, the manuscript is well written and clear, but some methodological issues need to be better clarified. Moreover, I found the discussion rather bulky and at times departing from the main aims and results of the work. I suggest focusing it more tightly and reducing its length.

In both the abstract and the discussion, the mention of the need for assisted colonization could be retained, but it does not require further detail since it is not a central topic of the study.

I agree with the reviewer about the need for more details regarding the exclusion of the 18 species with inconsistent data. What exactly does “low occurrence” mean? The species list should also include the author names for the scientific names to avoid any synonymy issues.

The exclusion of Natura 2000 areas is not well justified. The statement about “frequent overlap” does not mean that some areas could not be relevant to include in this work. Moreover, the mention of Plantlife Nature Reserves is unclear—were they included in the study or not?

It is also not clear what the authors mean in the following sentence:

“The decision to use data from this period of the plant atlas, despite newer records existing, aligns with Critchlow et al.'s [27] choice for their distribution models and reflects the greater likelihood that species’ distributions were in equilibrium with contemporary climate.”

The climate has dramatically changed over the past 40 years, as have potential species distributions. Perhaps the species’ distributions better reflect the period when the protected areas were established. I see here a possible weakness in the approach.

The section “Plantlife Nature Reserves” only confuses the reader; it should be merged with the other sections, specifying that the study was carried out in two steps. Also, merge “Analysis of climate refugia variables” with “Data analysis”.

Please carefully check the manuscript for typos.

Specific comments:

• L128: write the number in letters at the start of the sentence.

• L153–157: unclear whether these areas were considered here or not.

• L176–187: information about habitat suitability should be moved to a supplementary material table.

• L218: is the time span of the UKCEH Land Cover Map consistent with the plant occurrence data and the SDM maps?

• L248: remove the extra bracket.

• L340–360: the section about translocation can be summarized.

• L390: avoid using the term correlation when referring to model performance; use proper terminology instead.

Reviewers' comments:

Reviewer's Responses to Questions

**Comments to the Author**

1. Is the manuscript technically sound, and do the data support the conclusions?

Reviewer #1: Yes

2. Has the statistical analysis been performed appropriately and rigorously?

Reviewer #1: Yes

3. Have the authors made all data underlying the findings in their manuscript fully available?

Reviewer #1: Yes

4. Is the manuscript presented in an intelligible fashion and written in standard English?

Reviewer #1: Yes

Reviewer #1: After carefully reading the manuscript titled "Assessing protected areas as climate refugia for threatened plant species in Britain", I found this manuscript to be a timely and relevant contribution to the field of conservation biology under climate change, with a well-defined focus on identifying in situ and ex situ climate refugia for threatened plant species in Britain.

Overall, I find this manuscript scientifically sound, well-conceived, and valuable for guiding the conservation of threatened plants in Britain under climate change. However, some issues need to be addressed. The detailed technical review comment is listed as follows:

1) While the focus on 12 species allows for detailed modeling, the justification for excluding 18 species due to low occurrence could be elaborated, especially regarding how representative and comprehensive the focal species are for broader British plant biodiversity.

2) The discussion on microclimatic variation acknowledges its importance but relies on indirect inference due to model resolution. Integrating higher-resolution environmental heterogeneity or microclimate data would strengthen future versions.

3) The current habitat maps used to constrain future distributions may limit projections since vegetation and land cover will also respond to climate change. I suggest that potential impacts on refugia predictions should be discussed in greater detail.

4) The study area excludes Natura 2000 sites, considering their overlap with SSSIs, which may omit additional refugia or connectivity corridors. Including or at least discussing this limitation and its conservation implications would benefit the reader.

5) The longitudinal effect found contrasts with initial hypotheses and deserves a deeper ecological or climatological interpretation to clarify why eastern PAs support more species refugia.

6) Some terminology, such as "site suitability score" and threshold definitions, could be presented more clearly in methods for readers less familiar with SDM outputs.

7) Minor grammatical edits and language polishing would improve readability (e.g., shift between passive and active voice, occasional complex sentence structures).

Addressing the points above will further strengthen the manuscript’s clarity and impact.

**Do you want your identity to be public for this peer review?** For information about this choice, including consent withdrawal, please see our Privacy Policy

Reviewer #1: No

---

## [Author Response · Author response to Decision Letter 1]

10 Dec 2025

Dear Editor and Reviewer,

We would like to express our sincere gratitude for your valuable comments. We have considered all suggestions and have revised the manuscript accordingly. Below we have provided our responses outlining the changes we made to address each point.

Journal Requirements:

Response: Done. We have made multiple adjustments to ensure that the manuscript meets PLOS ONE’s style requirements, including changes to font size and the file names of the supplementary information. We apologise that these were not correct in the first submission.

2. Please amend your authorship list in your manuscript file to include author Freya Rebecca Read, Rachel Warmington, Colin Beale.

Response: Done

3. Please amend the manuscript submission data (via Edit Submission) to include author Freya R. Read, Rachel J. Warmington, Colin M. Beale.

Response: Done

Response: Done

5. We note that Figure(s) 1, S1 in your submission contains [map/satellite] images which may be copyrighted. All PLOS content is published under the Creative Commons Attribution License (CC BY 4.0), which means that the manuscript, images, and Supporting Information files will be freely available online, and any third party is permitted to access, download, copy, distribute, and use these materials in any way, even commercially, with proper attribution. For these reasons, we cannot publish previously copyrighted maps or satellite images created using proprietary data, such as Google software (Google Maps, Street View, and Earth).

For more information, see our copyright guidelines: http://journals.plos.org/plosone/s/licenses-and-copyright.

We require you to either (1) present written permission from the copyright holder to publish these figures specifically under the CC BY 4.0 license, or (2) remove the figures from your submission.

Response: The maps in Figure 1 and S1 use GADM data, which is freely available for academic use and other non-commercial use. It can be included in open-access academic research articles.

The GADM website states: "Using the data to create maps for publishing of academic research articles is allowed. Thus you can use the maps you made with GADM data for figures in articles published by PLoS, Springer Nature, Elsevier, MDPI, etc. You are allowed (but not required) to publish these articles (and the maps they contain) under an open license such as CC-BY as is the case with PLoS journals and may be the case with other open access articles. Data for the following countries is covered by a a different license Austria: Creative Commons Attribution-ShareAlike 2.0 (source: Government of Ausria)"

https://gadm.org/data.html.

A reference to GADM has been added in the methods section of the manuscript (lines 194-195 of the revised manuscript).

Additional Editor Comments :

In general, the manuscript is well written and clear, but some methodological issues need to be better clarified. Moreover, I found the discussion rather bulky and at times departing from the main aims and results of the work. I suggest focusing it more tightly and reducing its length.

Response: Thank you for the feedback. The discussion has been condensed, especially sections related to assisted colonisation.

In both the abstract and the discussion, the mention of the need for assisted colonization could be retained, but it does not require further detail since it is not a central topic of the study.

Response: Done. Multiple sentences on assisted colonization have been removed or condensed when they did not add to the central topic of the discussion. In the abstract the sentences on line 22-23 and 30-31 were removed and in the discussion the sections 340-361 and 525-528 were heavily condensed.

I agree with the reviewer about the need for more details regarding the exclusion of the 18 species with inconsistent data. What exactly does “low occurrence” mean?

Response: The following explanation has been added to the methods:

“We excluded eighteen species that were recorded in fewer than five hectads to ensure sufficient data for modelling (21,27).”

The species list should also include the author names for the scientific names to avoid any synonymy issues.

Response: Author names have been added to all species included in the paper in the first instance they appear (Line 119-130 in the revised manuscript).

The exclusion of Natura 2000 areas is not well justified. The statement about “frequent overlap” does not mean that some areas could not be relevant to include in this work.

Response: The following explanation has been added to the methods:

“The Natura 2000 sites that are not included in SSSI are predominantly marine areas, which are not relevant to this study (39,40).”

Moreover, the mention of Plantlife Nature Reserves is unclear—were they included in the study or not?

Response: Thank you for feedback on this section. The Plantlife Nature Reserves were not included in the main study. A small case study was carried out for Plantlife reserves using the same methods as the main analysis. In order to avoid confusion, this information of the Plantlife Nature Reserve work has been moved to a supplementary text file (S1 Text).

It is also not clear what the authors mean in the following sentence:

“The decision to use data from this period of the plant atlas, despite newer records existing, aligns with Critchlow et al.'s [27] choice for their distribution models and reflects the greater likelihood that species’ distributions were in equilibrium with contemporary climate.”

The climate has dramatically changed over the past 40 years, as have potential species distributions. Perhaps the species’ distributions better reflect the period when the protected areas were established. I see here a possible weakness in the approach.

Response: Thank you for your comment. The following explanation has been added to the methods section:

“Whilst climate has changed dramatically over the past 40 years, this data was collected during a period of high effort to record distributions across the UK, and precedes most of the modern climate change. This combination of quality data from a period when species were more likely to be in equilibrium with climate, makes our choice of dates appropriate.”

The section “Plantlife Nature Reserves” only confuses the reader; it should be merged with the other sections, specifying that the study was carried out in two steps.

Response: As addressed in comment 6 the Plantlife Nature Reserves case study has been moved to a supplementary file (S1 Text) to avoid confusion.

Also, merge “Analysis of climate refugia variables” with “Data analysis”.

Response: Done

Please carefully check the manuscript for typos.

Response: Done

L128: write the number in letters at the start of the sentence.

Response: Done

L153–157: unclear whether these areas were considered here or not.

Response: In order to avoid confusion, this information of the Plantlife Nature Reserve work has been moved to a supplementary text file (S2 Text). This case study was carried out separately and does not contribute to the main analysis of the paper.

L176–187: information about habitat suitability should be moved to a supplementary material table.

Response: Done. This has been moved to S1 Text.

L218: is the time span of the UKCEH Land Cover Map consistent with the plant occurrence data and the SDM maps?

Response: The time span of the UKCEH Land Cover Map is not consistent with the plant occurrence data and the SDM maps. An explanation for this has been added to S1 Text.

“The UKCEH Land Cover Map does not overlap with the period of the SDM. This is because the land cover maps were used after the SDM process as a spatial filter, to ensure that both current and future climate suitability are limited to habitat types that are currently present, based on the most recent land-cover information.”

L248: remove the extra bracket.

Response: Done

L340–360: the section about translocation can be summarized.

Response: Done. These two paragraphs have been summarised into a singular short paragraph (Line 294-300 in the revised manuscript).

L390: avoid using the term correlation when referring to model performance; use proper terminology instead.

Response: Done. The term “association” has been used instead.

Reviewer #1 Comments

1. While the focus on 12 species allows for detailed modeling, the justification for excluding 18 species due to low occurrence could be elaborated, especially regarding how representative and comprehensive the focal species are for broader British plant biodiversity.

Response: The justification for excluding the 18 species due to low occurrence has been expanded and a comment on how representative the remaining species are for broader British plant biodiversity. The following sentences were added to the introduction:

“We excluded eighteen species that were recorded in fewer than five hectads to ensure sufficient data for modelling (21,27). While this reduces the number of species analysed, the remaining 12 span a wide range of habitat types, levels of rarity and geographic distributions across the UK.”

2. The discussion on microclimatic variation acknowledges its importance but relies on indirect inference due to model resolution. Integrating higher-resolution environmental heterogeneity or microclimate data would strengthen future versions.

Response: The following section was added to the limitations section of the discussion:

“Incorporating higher-resolution environmental or topographic data in future analyses would strengthen these assessments. Tools such as climate data loggers, remote sensing and fine-scale digital elevation models can help identify microhabitats that contribute to local thermal buffering (Keppel et al., 2012), improving our understanding of how protected areas function as climate refugia.”

3. The current habitat maps used to constrain future distributions may limit projections since vegetation and land cover will also respond to climate change. I suggest that potential impacts on refugia predictions should be discussed in greater detail.

Response: The following section was added to the limitations section of the discussion:

“Additionally, in the UK large-scale transitions in land cover are more likely to be the result of a combination of human land use change and climate change. While integrated scenarios of climate and land use change are becoming available, they are not yet suitably detailed to enable modeling. Over the time periods we looked at, most models estimate more land covers will remain as they are rather than transition (77), making our results informative.”

4. The study area excludes Natura 2000 sites, considering their overlap with SSSIs, which may omit additional refugia or connectivity corridors. Including or at least discussing this limitation and its conservation implications would benefit the reader.

Response: The following sentence has been added to the paragraph in why Natura 2000 sites have not been included:

“The Natura 2000 sites that are not included in SSSI are predominantly marine areas, which are not relevant to this study (39,40).”

5. The longitudinal effect found contrasts with initial hypotheses and deserves a deeper ecological or climatological interpretation to clarify why eastern PAs support more species refugia.

Response: This paragraph has been expanded in order to go further into why eastern PAs support more species. The following has been added:

“This eastern pattern may partly reflect the particular species that passed the inclusion threshold, which are those with sufficient occurrence records to be reliably modelled. Many of these species are associated with arable land, calcareous grasslands, or inland rock habitats, which are more prevalent in eastern and southeastern Britain, creating a natural eastward bias in suitable habitat availability. Species meeting this threshold may also be biased toward regions with high observer effort. The southeast has historically had better coverage, especially around major towns and cities (20,39).”

6. Some terminology, such as "site suitability score" and threshold definitions, could be presented more clearly in methods for readers less familiar with SDM outputs.

Response: This paragraph has been adjusted in order to present the SDM results more clearly.

7. Minor grammatical edits and language polishing would improve readability (e.g., shift between passive and active voice, occasional complex sentence structures).

Response: Done. We have made edits throughout the manuscript. For example, we simplified complex sentence structure on lines 63-64, and converted several sentences from passive to active voice, most notably in the paragraph spanning lines 93-101.

We thank the Editor and the Reviewer for their time and constructive feedback on our manuscript.

Yours sincerely,

Freya R Read, on behalf of all co-authors

---

## [Editor Report · Decision Letter 1]

7 Jan 2026

Assessing protected areas as climate refugia for threatened plant species in Britain

PONE-D-25-47373R1

Dear Dr. Read,

We’re pleased to inform you that your manuscript has been judged scientifically suitable for publication and will be formally accepted for publication once it meets all outstanding technical requirements.

Kind regards,

Francesco Boscutti

Academic Editor

PLOS One

Additional Editor Comments (optional):

I have carefully revised the new version of the manuscript and confirm that the authors have adequately addressed all comments raised during the first round of review.

The manuscript requires only minor text corrections, which I have listed below:

L42: Remove “d”.

L82: “A combination of factors are likely” should be changed to “A combination of factors is likely”.

L143: Correct subject–verb agreement: “Data on the current plant distributions was” should be “were”.

L179: Correct subject–verb agreement: “Information on latitude, longitude and area were computed” should be “was”.

L216: Correct subject–verb agreement: “The spatial distribution of both in situ and ex situ refugia across Britain appear to have similarities” should be “appears”.

L434: Check reference formatting: “microhabitats that contribute to local thermal buffering (Keppel et al., 2012)” – earlier references are numerical; adjust to match the style.

L462: Correct reference placement: “Translocations can be used to relocate species to suitable habitats, a tool that will become increasingly important as global temperatures continue to rise. (2,78)” should be “…continue to rise (2,78)”.

Overall, the manuscript is now acceptable for publication following these minor adjustments.
---

## [Editor Report · Acceptance letter]

PONE-D-25-47373R1

PLOS One

Dear Dr. Read,

I'm pleased to inform you that your manuscript has been deemed suitable for publication in PLOS One. Congratulations! Your manuscript is now being handed over to our production team.

Kind regards,

on behalf of

Dr. Francesco Boscutti

Academic Editor

PLOS One